# WLAN RSS-Based Fingerprinting for Indoor Localization: A Machine Learning Inspired Bag-of-Features Approach

**DOI:** 10.3390/s22145236

**Published:** 2022-07-13

**Authors:** Sohaib Bin Altaf Khattak, Moustafa M. Nasralla, Maged Abdullah Esmail, Hala Mostafa, Min Jia

**Affiliations:** 1Smart Systems Engineering Lab, College of Engineering, Prince Sultan University, Riyadh 11586, Saudi Arabia; skhattak@psu.edu.sa (S.B.A.K.); mesmail@psu.edu.sa (M.A.E.); 2Communications Research Center, School of Electronics and Information Engineering, Harbin Institute of Technology, Harbin 150001, China; jiamin@hit.edu.cn; 3ACTSENA Research Group, Telecommunication Engineering Department, University of Engineering and Technology Taxila, Punjab 47050, Pakistan; engr.fawad@students.uettaxila.edu.pk; 4Department of Information Technology, College of Computer and Information Sciences, Princess Nourah bint Abdulrahman University, P.O. Box 84428, Riyadh 11671, Saudi Arabia; hfmostafa@pnu.edu.sa

**Keywords:** indoor positioning system, machine learning, WLAN fingerprinting, higher education, learning environment

## Abstract

Location-based services have permeated Smart academic institutions, enhancing the quality of higher education. Position information of people and objects can predict different potential requirements and provide relevant services to meet those needs. Indoor positioning system (IPS) research has attained robust location-based services in complex indoor structures. Unforeseeable propagation loss in complex indoor environments results in poor localization accuracy of the system. Various IPSs have been developed based on fingerprinting to precisely locate an object even in the presence of indoor artifacts such as multipath and unpredictable radio propagation losses. However, such methods are deleteriously affected by the vulnerability of fingerprint matching frameworks. In this paper, we propose a novel machine learning framework consisting of Bag-of-Features and followed by a k-nearest neighbor classifier to categorize the final features into their respective geographical coordinate data. BoF calculates the vocabulary set using k-mean clustering, where the frequency of the vocabulary in the raw fingerprint data represents the robust final features that improve localization accuracy. Experimental results from simulation-based indoor scenarios and real-time experiments demonstrate that the proposed framework outperforms previously developed models.

## 1. Introduction

Realizing the importance of evolving educational and pedagogical conditions, universities and colleges are aiming to include extensive upgrades and to develop exciting technological services [1,2,3,4]. The concept of Smart Campuses in Smart Cities seeks to integrate beneficial technological interventions such as localization and navigation, asset monitoring, smart attendance, and smart parking systems into the fabric of universities in order to improve learning, research, and managerial and administrative efficiency [5,6,7]. These applications require a robust positioning framework to precisely locate objects in indoor and outdoor environments [8,9]. A significant portion of the campus is disorganized, which can be difficult for not only students, but for faculty, staff, and particularly visitors. Students and visitors can use seamless indoor and outdoor positioning systems to navigate the campus. Outdoor location-based services can be easily achieved by employing global positioning system (GPS). However, in an indoor environment, non-line-of-sight communication, complex building structures, and other obstacles degrade the performance of satellite-based positioning. Thus, there is a strong need for reliable indoor positioning systems (IPSs). Indoor environments are usually highly cluttered, with many obstacles causing signal attenuation and creating blind spots that degrade the localization performance of the IPS [10]. Indoor positioning mechanisms identify the geographical coordinates of objects residing inside complex indoor structures. IPSs can be applied to track children and elderly users with wearable devices inside crowded malls and hospitals [11,12]. These positioning systems can also be used for military applications, mass rapid transit systems, or indoor areas where finding the location of a device user is mandatory.

Numerous approaches have been proposed for IPS without the aid of GPS. Such scenarios take help from anchor nodes. ANs already know their locations; they are either GPS equipped or manually deployed at points with known locations [13]. These localization techniques can be broadly classified into two categories: range-based and range-free [14]. Range-based localization works on radio transmission-based distance estimation, using these distances for position estimations. On the contrary, range-free localization algorithms use network connectivity, previous measurements, or other features that are not dependent on distance estimation [15]. These IPS frameworks mainly rely on three location-dependent parameters: angle, distance, and strength of the received anchor node signal [16]. Angle-based methods depend on the angle-of-arrival (AoA), which is estimated by the AoA of the incoming signals. Time-based methods, including time-of-arrival (ToA) and time-difference-of-arrival (TDoA), depend on source-to-destination signal propagation time. Strength-based methods depend on received signal strength (RSS).

AoA-based localization relies on the azimuth angle relative to the target nodes. AoA-based localization accuracy is highly vulnerable to non-line-of-sight multipath effects, in which reflected signals are received at a wrong azimuth angle and degrade the precision of the AoA method. Moreover, AoA requires complex hardware with proper calibration and synchronization for precise position estimation. The ToA method depends on the propagation time duration of the packet from the anchor node to the target node [17]. ToA-based indoor localization is less complex than the AoA method but is dependent on synchronization of the source and the destination node. The ToA value is extracted from the packet at the destination from the labeled timestamps [16]. Synchronization requires the integration of more hardware in the indoor positioning system, which increases the cost of the IPS. The TDoA framework’s positioning accuracy is also sensitive to synchronization of the anchor and target nodes. In RSS-based indoor positioning, objects are localized by the strength of the signal at the receiving target node [18]. In our work, we choose RSS for position estimation as it is the simplest method and does not require additional hardware for synchronization. The reason for the wide popularity of RSS is that almost all radio-enabled devices can process and display RSS.

When it comes to indoor positioning, range-based techniques have a drawback: they are badly affected by multipath fading and shadowing in cluttered indoor environments. Due to these factors, variability in the values can cause large estimation errors. In comparison, the range-free techniques do not consider these mathematical models and are not prone to error compared to range-based techniques. For these reasons, range-free techniques are preferred indoors, some examples of range-free positioning techniques are cell-identity-based, proximity-based, fingerprinting, etc. Fingerprinting is a range-free technique for localization, and it can be applied in all wireless communication systems. A fingerprint is a unique set of location-dependent signal parameters, so each location has a unique fingerprint associated with that location. It is the most reliable method for localization in RSS-based indoor positioning systems. Fingerprinting is comprised of two stage: offline training and online testing. In the offline training stage, the RSS values of the anchor nodes are recorded at each reference point (RP), a geographic coordinate in the coverage area. During the online testing stage, the RSS values at unknown coordinates are subjected to a robust classifier that estimates the position’s coordinate based on the training data [19]. Although fingerprinting-based IPSs are considered reliable, they face some challenges because of variations in permittivity and permeability of materials in the signal propagation path, creating nonuniform propagation loss. Moreover, the multipath effect in indoor environments also threatens the signal strength at target node positions. Various positioning algorithms using a number of classification models have been adopted to estimate the precise 2D coordinates. These models categorise the observed training data and assign the test data to the best set. The most reliable classification models include the multi-class support vector machine (SVM), K-nearest neighbor (KNN), decision tree (DT), and the ensemble classifier. The classification accuracy of these classifiers depends on the RSS collected during the offline training stage. Moreover, the robustness and distinctiveness of the fingerprint data provide more accuracy in the IPS framework. The data collected during offline training, consisting of RSS from each anchor node, is deficient due to cost constraints on the APs in the finite geography. The limited fingerprint training data at each coordinate affects the performance of the overall IPS framework. To handle this issue, we adopt a pre-processing Bag-of-Features (BoF) strategy that improves the overall strength and robustness of the training data.

In this paper, the proposed approach transforms the raw data into a high-dimensional form and makes it more compatible with the pre-existing classification models. The proposed approach formulates positioning as a pattern-recognition problem, where for each location a featured vector is obtained using a simplified BoF-based technique. It consists of characterizing several WLAN RSS measurements observed at each RP. The BoF model is applied to the raw RSS fingerprint data by accumulating k-mean clustering and collecting the differential vectorization of vocabulary set occurrences. This is the first time that such a technique has been used in WLAN RSS fingerprinting-based indoor localization. Previously, researchers employed pre-processing methods such as spectrogram transformation [20] and interpolation [21]. However, such methods extend the complexity of the model by increasing the data size. In this work, the proposed approach is tested in different simulation scenarios, and a real time experiment is also conducted to validate the proposed strategy. The proposed framework is compared with previously reported work, and the overall results testify of the superiority of our model.

The rest of the paper is organized as follows. Section 2 briefly reviews the literature relevant to the considered problem. Section 3 describes the system model, and Section 4 includes the proposed approach. Section 5 presents the simulation results and the real-time experiment; and finally, Section 6 is the conclusion.

## 2. Related Work

The existing literature focuses on efficient indoor positioning techniques that are low cost and are accurate in diverse environments. An ideal IPS would work in numerous indoor scenarios. To analyze the best indoor localization framework, a brief comparative investigation of the literature is presented. Existing research shows that machine learning models benefit IPS frameworks both in terms of cost and precision. Extensive research on handcrafted and deep learning models has assisted in achieving noise robustness in indoor positioning systems [22]. With the development of advanced computational devices, the employment of deep neural models and advanced machine learning frameworks has become possible [23,24]. Deep neural networks such as AlexNet, DagNet, GoogleNet, ResNet, InceptionV3, VGG-16, MobileNet, and ZFNet require 2D input data, which is not available in the case of fingerprint base localization [25]. Indoor positioning systems developed for tracking objects broadly fall in two main classes: wireless signal and vision frameworks. Vision-based frameworks employ computer vision algorithms on the images captured by mounted cameras serving in indoor environments. Vision-based methods detect the desired object in non-overlapping camera networks of buildings by classifying the robust visual features extracted from the region of interest [26]. Vision-based methodologies are highly precise but computationally expensive. The high frame-rate and extensive resolution of images requires expensive processors and graphics processing unit for the recurrent neural networks to recognize and annotate the desired object in the input camera feeds. Moreover, vision-based frameworks also suffer from geometric and photometric variations, including occlusions and illumination and viewpoint variations. Compared to vision-based methods, wireless signal-based methods are cost-effective and require fewer computational resources [27].

Wireless signal-based indoor positioning frameworks rely both on geometric and fingerprinting methods. The method developed in [28,29] considers indoor localization as a Gaussian and KNN regression problem. Regression-based models have less complexity, but they focus only on classification of pre-existing fingerprint data and ignore robust feature extraction. The work in [30] considers the importance of features to improve the precision of indoor localization by including continuous wavelet transforms (CWT) on raw RSS fingerprint data. CWT converts the 1D vector RSS data into 2D image data, with which pre-existing deep neural network models are easily compatible. CWT transformation is an additional stage before feature extraction that improve precision at the cost of the computational complexity of the model. Other works can be found in the literature using principal component analysis for feature extraction [31,32].

Deep neural network (DNN) frameworks are much more sensitive to the format of input data, whereas RSS fingerprint data is vectorized data with a limited set of values for each geographical coordinate. Therefore, most IPSs incorporate handcrafted statistical models in their machine learning frameworks for indoor localization. Handcrafted statistical models are less complex and embrace the problem’s specific modifications. The authors in [33] integrated a statistical hypothesis test on asymptotic relative efficiency (ARE) to optimize signal distribution at the site coverage area. Another work [34] introduces multi-output least square support vector machine (M-LS-SVM) regression to improve classification of RSS fingerprint data. Localization in [35] is achieved by fusion of grid-independent and grid-dependent least-square classifications. The authors of [36] used a neural network-based algorithm to correct the camera tilt angle, and they used nueral networks to establish a relationship between LED images and distance. However, the noise generated by reflection, which affects IPS performance, has not been considered. In this work, we focus on the robustness and distinctiveness of RSS fingerprint data with the support of pre-processing to make pre-existing indoor localization frameworks more efficient and accurate. Previously reported frameworks mainly focused on enhancement of RSS fingerprint data by suppressing noise through both pre-processing and post-processing, whereas our proposed approach transforms the raw fingerprint data into a more meaningful shape that is robust and leads to highly accurate localization. BoF and Bag-of-Words (BoW) [37] exist for image and document classification, respectively, whereas here it is implemented and incorporated for RSS-based indoor positioning for the first time.

## 3. System Model

Fingerprint-based localization systems are divided into two phases: offline and online. Figure 1 depicts the WLAN fingerprinting-based indoor positioning system. A radio map is created offline by dividing the area of interest into grids or RPs. At these RPs, a survey is conducted to collect RSS readings from the accessible APs, and then a database is produced, as illustrated in Figure 1. This database is the radio map, which contains map-like identification features but is based on the RSS of the radio waves. The signature created at each RP serves as the RP’s fingerprint. On the other hand, during the online phase, the user initiates a query from a specific point inside the area of interest. The system uses different matching algorithms to compare the query with the radio map, and then the most comparable fingerprint is returned as the estimated position. The RPs can be mathematically represented as follows
(1)$$RP_{j}=\left(x,y\right),j=\left(1,.....N\right),$$
where (x,y) is the coordinate point of the RP in the grid-based area, and *N* is the total number of RPs. The fingerprint database or radio map can be mathematically described as follows
(2)$$\lambda =\left[\begin{array}{cc} RP_{1} & \left(\psi_{1,1},..,\psi_{1,M}\right)\\ : & :\\ RP_{N} & \left(\psi_{N,1},..,\psi_{N,M}\right) \end{array}\right]$$
where $\psi_{i,j}$ refers to RSS samples at *i*th AP from *j*th RP, and *M* is the total number of APs.

To simulate signal transmission over the channel between the AP and RP, a log-normal path loss model is employed. This model can be used for a wide range of environments and considers the random shadowing effects caused by different types of obstacles causing signal blockage. Shadowing effects cannot be ignored when modeling real environments. In indoor situations, path loss is affected by a variety of parameters, including distance (*D*), noise (ζ), physical barriers (γ), and human presence (ρ). Each barrier, whether a wall or a human, must have the resulting attenuation represented in the model. As a result, we use the extended log-normal path loss model in our simulations [38].
(3)$$P\left(d\right)=P\left(d_{0}\right)+10\cdot N\cdot log\left(d/d_{0}\right)+\zeta +\sum_{\iota =1}^{\upsilon }\left(\gamma_{\iota }\right)+\sum_{\kappa =1}^{\Upsilon }\left(\rho_{\kappa }\right)$$
where P(d) denotes the RSS at point *d* in the (x,y) coordinate system, $P(d_{0})$ denotes the RSS at reference distance (1 m), N denotes the path loss coefficient, ζ denotes the shadowing effect, and *d* denotes the distance between AP and RP. In the summation for wall and human attenuation factors, ι is the ιth physical barrier (walls in particular), υ represents the total number of barriers, and κ is the κth human in the path, with Υ as the total number of humans through which the signal attenuated. The RSS values fluctuate over time due to many factors that contribute to signal fading. To counter temporal variations, RSS readings are taken over a period of time, which is defined mathematically by
(4)$$RSS\left(i,j\right)=(S_{i,j}\left(\tau 1\right)\ldots ..S_{i,j}\left(\tau \Gamma \right))$$
(5)$$\psi_{i,j}=\frac{\sum_{\tau =1}^{\xi }S_{i,j}\left(\tau \right)}{\Gamma }$$
where *i* is the *i*th AP, *j* is the *j*th RP, $S_{i,j}\left(\tau \right)$ is the RSS sample collected at time instant τ, and Γ is the total number of collected RSS samples. The average value of RSS samples is used in the fingerprint database.

## 4. Methodology

In the proposed framework, a BoF model is introduced for pre-processing to achieve noise robustness and distinctiveness in the feature accumulation process of indoor positioning based on RSS fingerprinting. The proposed BoF approach employs RSS fingerprint training data for vocabulary generation based on clustering. The vocabulary here is used to create the feature vectors employed both in training and testing of the classification model. Figure 2 shows how the proposed approach is different from conventional fingerprinting algorithms. RSS fingerprints consist of the coordinates of the RP and the raw RSS values corresponding to each AP. The BoF framework considers the geographical coordinates as labels but the RSS from each AP as raw features.
(6)$$X=\left[\begin{array}{c} RP_{1}\\ :\\ RP_{N} \end{array}\right]$$
(7)$$Y=\left[\begin{array}{c} \left(\psi_{1,1},..,\psi_{1,M}\right)\\ :\\ \left(\psi_{N,1},..,\psi_{N,M}\right) \end{array}\right]$$
where *X* in Equation (Equation 6) represents the location coordinates of RPs, while $\psi_{i,j}$ in Equation (Equation 7) denotes the RSS value of the *i*th AP at *j*th RP. The BoF generates clusters by employing the raw feature set *Y* and records the cluster centers as vocabulary features.
(8)$$\begin{array}{cc} L=\sum_{v=1}^{k}\sum_{i=1}^{n}a_{<i,v>}{\left\{∥\psi_{i,j}-\mu_{i}^{v}{∥}^{2}\right\}}_{i=1}^{M}\; & \mathrm{where}\; \end{array}a_{<i,v>}=\left\{\begin{array}{cc} 1, & \mathrm{if}\;\psi_{i,j}\;\mathrm{belong}\;\mathrm{to}\;v\\ 0, & \mathrm{otherwise} \end{array}\right.$$
where L in Equation (Equation 8) denotes the cost function, and $\mu_{i}^{v}$ is the *v*th cluster center of the *i*th AP RSS. The variable μ denotes the cluster mean value, which updates with each iteration. The coefficient *a* in Equation (Equation 8) defines the class of each ψ vector associated with each grid point. The variable *v* denotes the cluster class label in the group of k classes. The k-mean clustering minimizes L with respect to *a* and μ, as given bellow:(a)Initialize $\mu_{i}^{1}$ to $\mu_{i}^{k}$ arbitrarily;(b)Choose the optimal *a* for a fixed μ;(c)Choose the optimal μ for a fixed *a*;(d)Repeat steps (b) and (c) until convergence.

The BoF model vocabulary set $\mu_{1}^{v}$ with *v* ranges from [1K], with a total of *K* cluster centers used to create a final feature vector associated with each RP of the known training and the required test data. The final training data *T* consisting of the final feature $\tilde{Y}$ and location labels *X* is used to estimate the location based on the test RSS value at an unknown location of the serving region.

In Equation (Equation 9), the final feature vector $\tilde{Y}$ from the raw RSS data *Y* is calculated for each reference point based on the available vocabulary set $\mu_{i}^{v}$. The mathematical form is given by
(9)$${\tilde{Y}}_{i}={\left\{{\left\{\psi_{i,j}-\mu_{i}^{v}\right\}}_{v=1}^{K}\right\}}_{i=1}^{M}.$$

The training data consisting of the final feature vector and its associated labels $<\tilde{Y},X>$ is used to train the classifier to estimate location coordinates based on the test data. The BoF model extracts the feature vector from the raw data, which remain robust and distinct in the presence of both spatial and spectral noise. The test feature vectors are classified with a trained classifier, and the estimated location is compared with the true location to calculate mean indoor positioning error.

The test features of the BoF models are classified with the distance-based classifier. Using KNN classification provides better precision when integrated with the proposed BoF model. The KNN method identifies the K nearest feature vectors in $\tilde{Y}$ with the lowest Euclidean distance *d* to the existing training features. For a test feature ξ in $R^{M}$ vector space, the Euclidean distance is shown in Equation (Equation 10).
(10)$$d_{i}=\sqrt{{\left(\xi_{j}-\psi_{i,j}\right)}^{2}}$$
where $d_{i}$ in Equation (Equation 10) represents the distance of test feature ξ and *i*th feature vector $\tilde{Y}$ from the training data. The test ξ is assigned to the majority vote of the k-nearest features of the training data $N_{k}\left(\tilde{Y}\right)$ shown in Equation (Equation 11). The $N_{k}\left(\tilde{Y}\right)$ are the *k*-nearest features of the test feature ξ in the training set $\tilde{Y})$. The optimal value of k in the KNN algorithm is used differently in the literature [39,40]. Usually, the value of K is between 3 and 5; for larger values the accuracy of the system degrades, as outliers are also included as neighbors.

Fractional probability *p* is used to assign class label *X* to test feature ξ.
(11)$$p\left(X_{i}/\xi \right)=\frac{1}{k}\left(\sum_{i\epsilon N_{k}\left(\tilde{Y}\right)}I\left(X==i\right)\right)$$

The class label, i.e., the RP coordinates, is assigned to test feature ξ on the basis of the fractional probability given in Equation (Equation 11). The class label for which $p(X_{i}/\xi )$ results in a maximum value is assigned to each test feature during the experiment. The KNN classification model is less complex; hence, it is integrated with the proposed pre-processing approach. The step-by-step functionality of the proposed approach can also be shown in Algorithm 1 and Figure 3.
**Algorithm 1:** Syntax of the proposed BoF indoor positioning model.
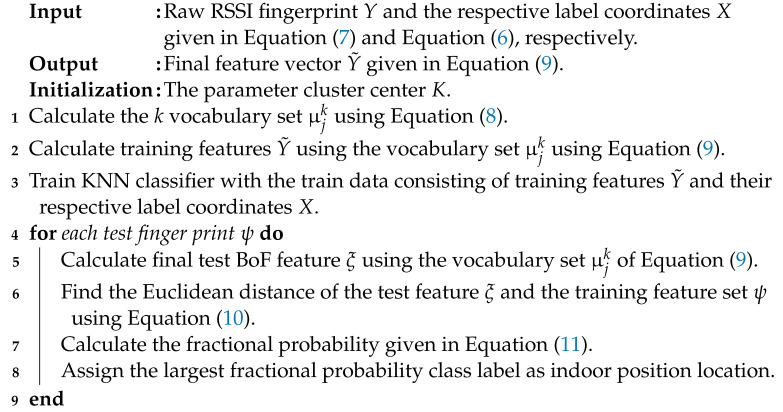


## 5. Performance Evaluation

In this section, we present a comprehensive performance evaluation of the proposed BoF-based positioning approach, which reveals the positioning accuracy. The experimental results of the proposed BoF indoor positioning model are validated through both simulated virtual and real-time testbeds. We choose two simulation scenarios from our previous work [41,42], and a personal testbed was set up consisting of two apartments in a residential building. All approaches are run on MATLABR2018a installed on a Dell laptop equipped with an Intel Core i7 processor and 8 GB RAM. Classification of raw RSS data features is carried out through KNN [39,43], probabilistic [44], SVM [45], DT [46], ensemble learning-based classification model [47], and the discriminant analysis classifier (DAC) [48], and comparisons are made with the proposed BoF-enabled approach. These are the most common machine learning models used in RSS fingerprinting systems [49]. The KNN method estimates labels based on k neighbor samples in the data. SVM separates test data into two categories based on the hyperplane. DAC classifies raw RSS data based on Gaussian distribution. The probabilistic model classifies test data based on the likelihood of training data samples. Ensemble learning uses multiple decision tree models and estimates the test sample class based on majority voting. However, the best results are obtained by integrating KNN classification with the proposed BoF feature extraction approach.

To calculate positioning error, Euclidean distance is calculated between estimated position $P_{Est}$ and actual position $P_{Test}$ of the test points (TPs). The overall localization accuracy of the system can be given by two performance metrics: mean absolute error (MAE) and cumulative distribution function (CDF) [40]. Let $N_{TP}$ be the number of TPs; MAE is defined as
(12)$$MAE=\frac{1}{N_{TP}}\sum_{\alpha =1}^{N_{TP}}\sqrt{{\left(P_{Test}-P_{Est}\right)}^{2}}$$

The CDF plot displays that the probability of positioning error (PE) is equal to or less than a certain distance. It shows the spread of the positioning errors of TPs and presents the comparison in terms of the proposed approach with various reported methods that have already been mentioned.

### 5.1. Virtual Environments

The model is validated in a virtual environment including walls and humans. The simulation environments resemble typical application scenes with differently sized rooms and corridors. The first virtual environment has the dimensions of 30 m × 30 m with four APs (red squares), while the second simulation scenario has the dimensions of 50 m × 50 m with five APs (red squares), as shown in Figure 4 and Figure 5, respectively. Simulations are performed to test the proposed algorithm in realistic and complex indoor environments; both scenarios include multiple rooms, corridors, and halls to match a typical university setting. Random human presence has also been included in both scenarios. In the first scenario, each installed WLAN AP covers the complete area of interest; some areas may have low signal quality. The second scenario considers a large indoor environment, where each AP cannot provide coverage to all RPs. Their coverage is restricted to a certain number of RPs where they can provide localization services. Hence, only a set of installed APs participate in localization. Both simulation scenarios have a grid spacing of 2 m. The RSS values to each RP have been assigned by the extended path loss model with human [50] and wall attenuation factors [51], and 20 samples are observed at each RP from all APs. The WLAN AP configurations for both simulation scenarios can be seen in Table 1, and the simulation parameters are listed in Table 2.

In the offline stage, BoF features are generated by using cluster dimensions of size 2, which leads to a feature dimension of 10 variables for the considered environment of 5 APs. In the online stage, RSS values from the available WLAN APs are initially translated through BoF, and then the KNN classification algorithm is applied to estimate the indoor location of the test device. For comparison, we employ simple KNN, probabilistic, SVM, DT, ensemble learning-based classification, and DAC, with the proposed BoF-assisted KNN outperforming comparatively.

The virtual 30 m × 30 m environment is tested using the BoF-based indoor positioning approach. The proposed BoF approach provides an MAE of 1.702 m, which is lower than that of the other models. The MAEs of KNN, probabilistic, SVM, DT, ensemble learning-based classification, and DAC are shown in Table 3 and are higher than our proposed BoF approach. The CDF plot of this simulated environment is shown in Figure 6.

In addition, the proposed BoF approach is validated on a virtual environment of dimensions 50 m × 50 m, resulting in an MAE of 2.837 m, which is approximately 1.1 m less than the second-best KNN classification model. The MAEs of the other methods are shown in Table 4. Moreover, the CDF plot of this environment is shown in Figure 7. It can be noted that the CDF plot shows our model outperforms previously reported methods.

### 5.2. Real-Time Testbed Experiment

All experiments are performed inside a residential building between two adjacent apartments. These two apartments have a living room and a bedroom each. The floor map of the area can be seen in Figure 8. This area is divided into two regions, namely, Region A and Region B, with dimensions of 3.95 m × 11.1 m and 8.4 m × 9.3 m, respectively. In this environment, we label a total of 30 RPs for the site survey, in which 10 RPs are in Region A and 20 RPs are in region B, with grid spacing of two meters, as shown in Figure 9. This experimental environment is less crowded and remains the same throughout the day, with no significant change. A Huawei smartphone (model: KIW-L21) with an android application installed [52] is used to collect Wi-Fi RSS data from five TPlink A1200 APs (notated as AP1 to AP5) for 25 s at each RP with 20 RSS samples. The details of this environmental setup along with photos of the hardware devices used for the real-time experiment can be seen in Figure 10. In a similar way, 10 TPs are also labeled for testing, where 3 TPs belong to Region A, and 7 TPs belong to Region B. The MAE obtained by BoF in the real environment is 1.581 m, which is 0.19 m lower than that of the KNN model. The MAEs of KNN, probabilistic, SVM, DT, ensemble learning-based classification, DAC, and the proposed model are given in Table 5, and the CDF plot can be seen in Figure 11.

## 6. Conclusions

The indoor positioning framework presented in this article enhances the reliability of positioning accuracy. The proposed BoF model transforms the raw RSS of the access points into robust and distinctive features with reduced localization error. Experimental validation of the proposed BoF integrated with KNN tested on both virtual and real-time testbeds shows promising performance. The proposed approach scores 1.702 m, 2.837 m, and 1.581 m mean absolute error on the simulated 30 m × 30 m and 50 m × 50 m and the real-time residential apartment environment, respectively, indicating lower error than other methods. Moreover, the CDF graphs clearly show the performance of our proposed approach remains more robust and distinct than state-of-the-art models, even in the presence of environmental artifacts. Machine-learning based pre-processing integrated with the simplest of classifiers can outperform conventional classification models and overcome their limitations. In future work, we will explore other ML approaches that can be integrated into the conventional IPS framework to enhance performance in complex indoor scenarios with limited training data.

## Figures and Tables

**Figure 1 sensors-22-05236-f001:**
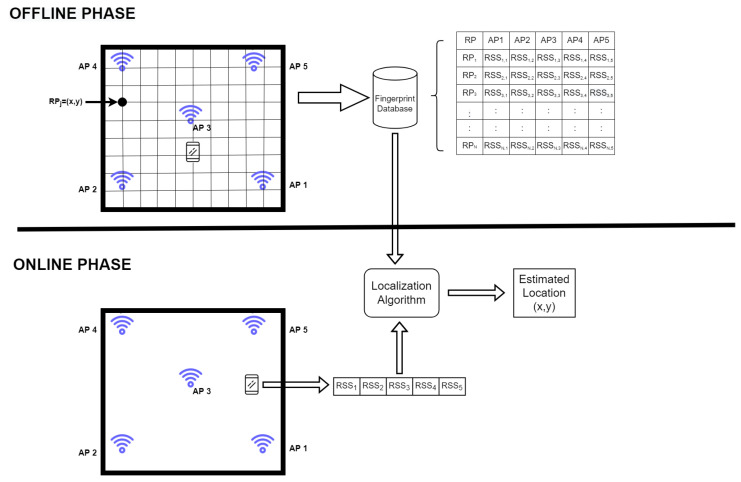
WLAN fingerprinting-based indoor positioning system.

**Figure 2 sensors-22-05236-f002:**
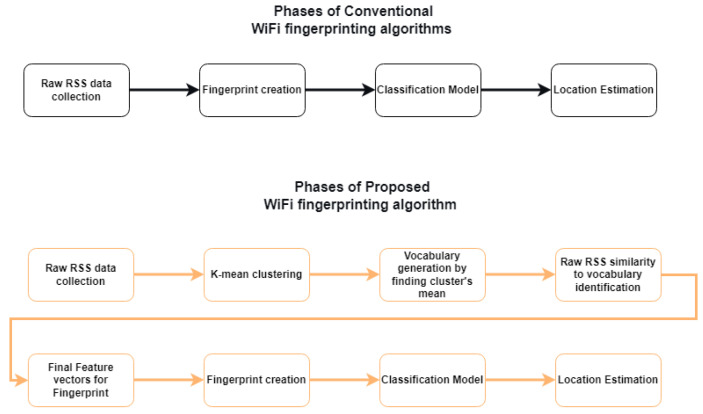
Flowchart of conventional fingerprinting-based IPS and proposed BoF-assisted approach.

**Figure 3 sensors-22-05236-f003:**
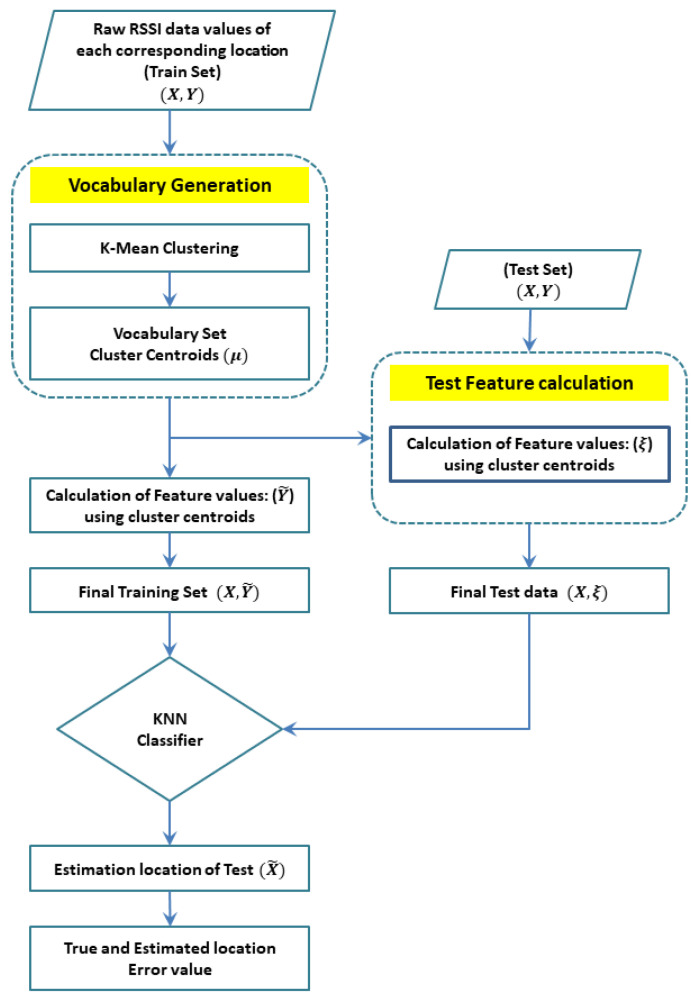
Step-by-step functionality of the proposed BoF-assisted approach.

**Figure 4 sensors-22-05236-f004:**
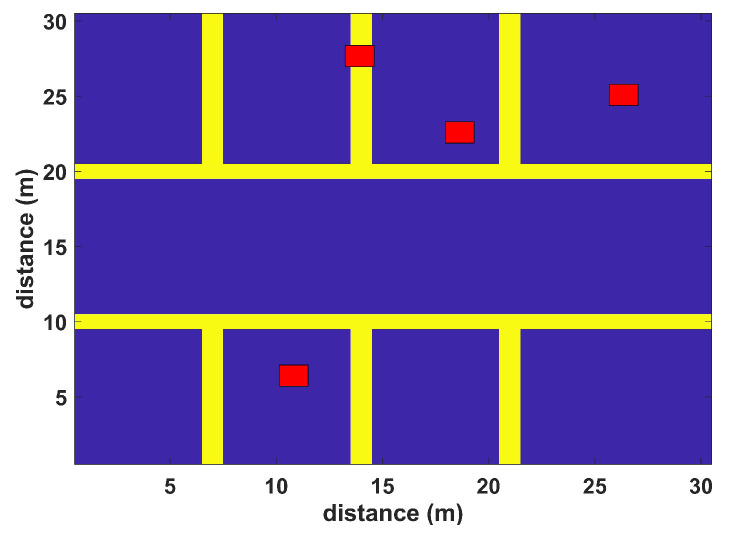
Floor map with 30 m by 30 m dimensions.

**Figure 5 sensors-22-05236-f005:**
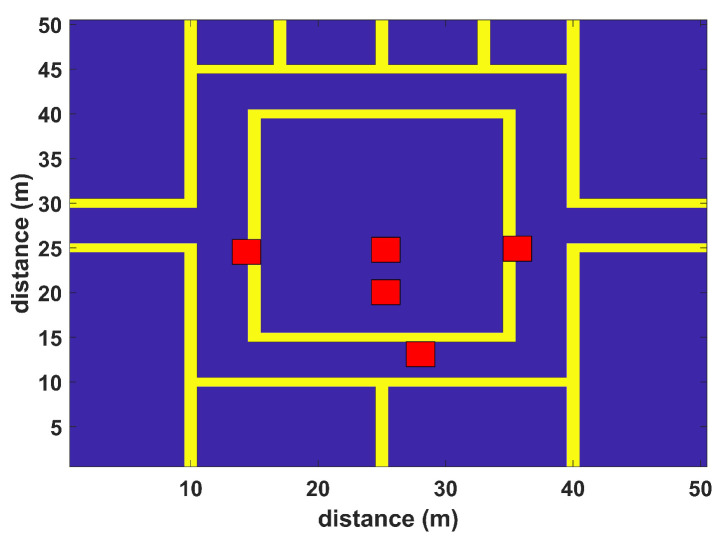
Floor map with 50 m by 50 m dimensions.

**Figure 6 sensors-22-05236-f006:**
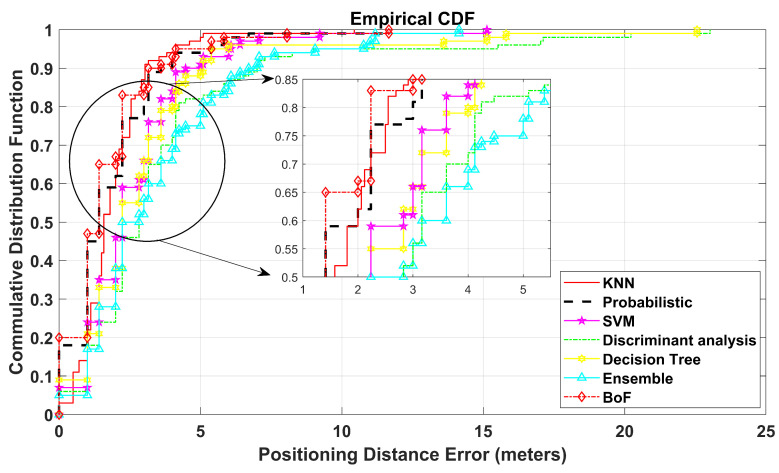
CDF plot for floor with 30 m × 30 m dimensions.

**Figure 7 sensors-22-05236-f007:**
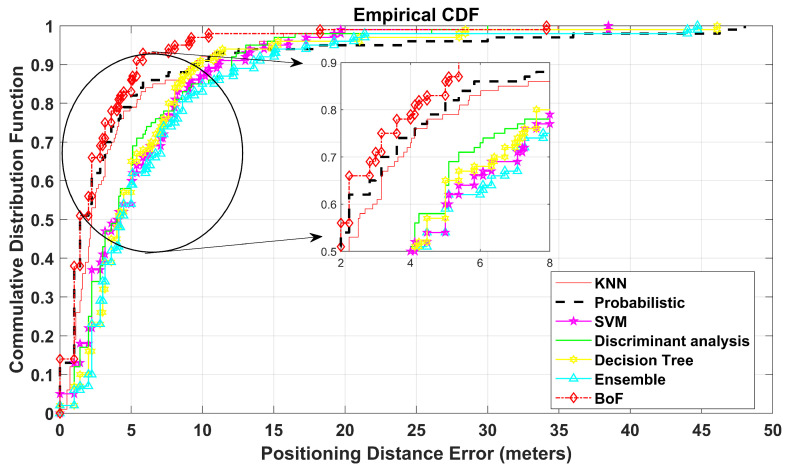
CDF plot for floor with 50 m by 50 m dimensions.

**Figure 8 sensors-22-05236-f008:**
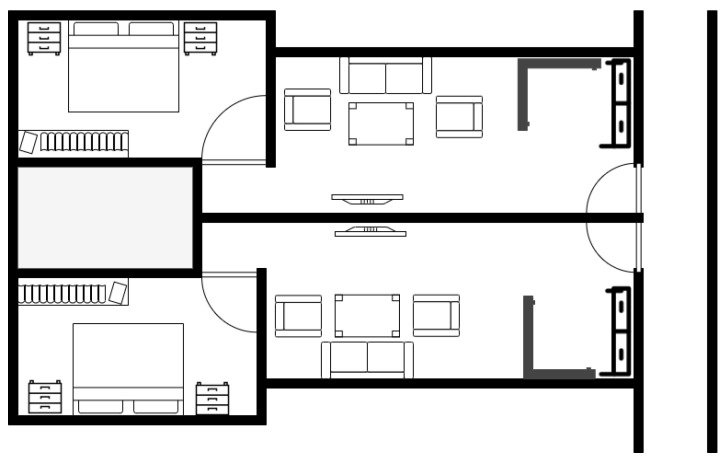
Experimental area floor map.

**Figure 9 sensors-22-05236-f009:**
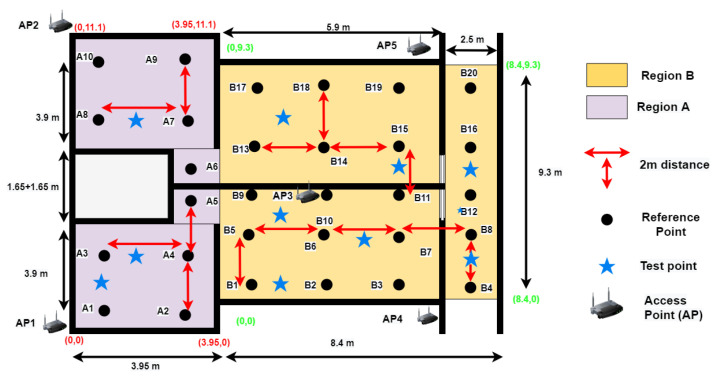
Experimental area radio map construction.

**Figure 10 sensors-22-05236-f010:**
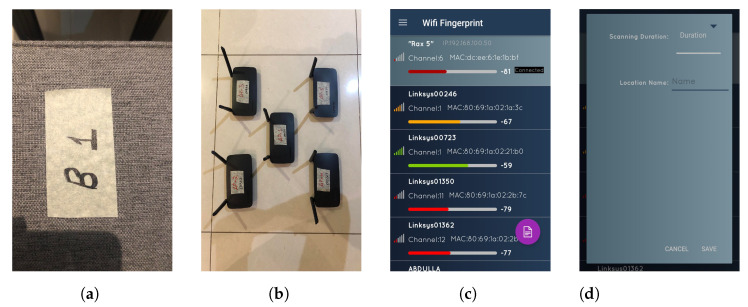
Basic elements of the fingerprinting experiment. (**a**) Grid mark; (**b**) APs used in the experiment; (**c**) Fingerprint utility home screen; (**d**) Fingerprint utility RSS sample collection.

**Figure 11 sensors-22-05236-f011:**
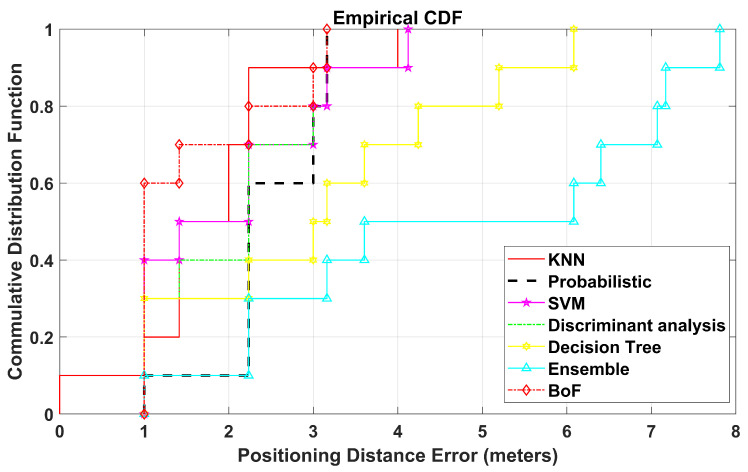
CDF Plot for real-time experiment.

**Table 1 sensors-22-05236-t001:** WLAN access point configurations.

Scenario	Dimensions	APs	Coordinates of APs
Scenario 1	30 m × 30 m	4 APs	(25, 25) (19, 23) (13, 27) (11, 7)
Scenario 2	50 m × 50 m	5 APs	(25, 25) (37, 25) (25, 19) (28, 13) (16, 25)

**Table 2 sensors-22-05236-t002:** Parameters used in simulation.

Parameter	Value
Path loss exponent *n*	3
Number of APs	4 and 5
Wall attenuation factor	4 dB
People attenuation factor	3 dB
Reference distance $d_{0}$	1 m
Power at $d_{0}$	−30 dBm
Transmission power	10 dBm
RSS samples collected at RP	20
*k* in KNN	4
Grid size	2 × 2
No. of position queries in virtual environments	100

**Table 3 sensors-22-05236-t003:** Mean absolute error of 30 m by 30 m floor.

KNN	Probabilistic	SVM	Discriminant Analysis	Decision Tree	Ensemble Learning	Bag-of-Features
1.922	1.841	2.617	3.729	3.438	3.005	1.702

**Table 4 sensors-22-05236-t004:** Mean absolute error of 50 m by 50 m floor.

KNN	Probabilistic	SVM	Discriminant Analysis	Decision Tree	Ensemble Learning	Bag-of-Features
3.929	4.447	5.480	5.298	6.531	5.820	2.837

**Table 5 sensors-22-05236-t005:** Mean absolute error in the real environment.

KNN	Probabilistic	SVM	Discriminant Analysis	Decision Tree	Ensemble Learning	Bag-of-Features
1.772	2.450	2.017	2.029	3.053	4.678	1.581

## Data Availability

Not applicable.

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
