# Peer review of "WLAN RSS-Based Fingerprinting for Indoor Localization: A Machine Learning Inspired Bag-of-Features Approach"

_sensors, 2022, doi:10.3390/s22145236_

Round 1
Reviewer 1 Report
o This propose a novel machine learning framework consisting of Bag-of-Features and followed by a k-Nearest Neighbor classifier.
o For the "smart cities" in the title. The proposed framework is within the scope of traditional WiFi finger print approaches. I couldn't find any special things for "smart cities".
o The experiment is simple and all parts have similarities to those found in previous studies. It is difficult to say that it is particularly advanced from related work.
o However, authors is proposing "framework" . It is easy for readers to read and implement it by themselves. This paper will be read a lot in that repects, and will provide many advantages.
Author Response
Please find the responses in the attached file.

Reviewer 2 Report
The research is focused on Lcalization Problem for indoor systems.
It has so many applications for childern and or elder in the case of crouded mall and or hospitalls.
I think the combination between fingreprints and RSS measurment is adding some sort of customer identification with location.
Verfication with other previously published works will justify the presented methdeology and its infulence in the indoor localization techniques. Also, flow chart to illustrate trhe presented procedure.
The error is so huge, so I am rcommending to minmize the error via some sort of mathematical manpulations and estimation process.
The references are appropriate but needs some additional refreences for similar papers treating the same problem.
In the conclusion part, you said "These scores have been observed as a lower error in comparison to the relative methods."
But practically no correction methodology has been used to minimize the error ?
The results need to be compared to another technique such as https://doi.org/10.3390/s21030719
Author Response

(The authors gave the same response as above.)

Round 2
Reviewer 2 Report
Results may be compared in order to get more perfection specially, in the indoor applications